# Development of a Phenology Model for Egg Hatching of Walking-Stick Insect, *Ramulus mikado* (Phasmatodea: Phasmatidae) in Korea

**Min-Jung Kim** [1] , **Keonhee E. Kim** [1] , **Seulki Son** [2] , **Yonghwan Park** [1], **Youngwoo Nam** [1,*] and **Jong-Kook Jung** [2,*]

1   Forest Entomology and Pathology Division, National Institute of Forest Science,
    Seoul 02455, Republic of Korea; minjungkim1022@gmail.com (M.-J.K.); kheekimeg@gmail.com (K.E.K.);
    parkyonghwan@korea.kr (Y.P.)
2   Department of Forest Environment Protection, Kangwon National University,
    Chuncheon 24341, Republic of Korea; sealki5686@naver.com
*   Correspondence: orangmania99@korea.kr (Y.N.); jkjung@kangwon.ac.kr (J.-K.J.)

**Abstract:** The walking-stick insect *Ramulus mikado* is occasionally considered a forest pest, as its mass occurrence can cause severe defoliation. It overwinters as eggs on the ground surface, and the hatched nymphs climb up to the host trees in spring. In this study, temperature-dependent development experiments were performed on *R. mikado* eggs under three constant temperatures (23.3, 28.3, and 29.2 °C) to extend the previously reported thermal response. The development times of eggs collected in summer and winter were also compared to investigate how development status is influenced by the seasons. The lower and upper developmental thresholds and thermal constants of *R. mikado* eggs were estimated to be 6.1 °C, 29.2 °C, and 1707.8 DD, respectively. The starting point for effective temperature effects on the eggs was estimated to be 1 August, based on the results of the experiment on field-collected eggs. A phenology model was constructed by using a development completion model scaled by the thermal constant, with a starting point of degree-day accumulation. The model showed good agreement, with a deviation of 3.2 ± 2.95 days between prediction and observation. The developed phenology model is useful for determining the appropriate timing for management decision-making regarding this insect.

**Keywords:** *Ramulus mikado*; walking-stick insect; egg hatching; degree-days model; forest pest; phenology

## 1. Introduction

The parthenogenetic walking-stick insect *Ramulus mikado* (Rehn, 1904), also known as *Baculum elongatum* or *R. irregulariterdentatus*, is native to East Asia, including China, Japan, and Korea [1,2]. With a long and slender body that is typically brownish-green in color, *R. mikado* is capable of camouflaging itself among twigs and branches [3]. This species is wingless, slow-moving, and nonaggressive toward people, making it relatively easy to handle.

The population of *R. mikado* has been maintained at a low level during the past several decades, and the insect has not had a high level of public awareness. However, instances of foliar damage caused by this insect on a variety of broadleaf trees, such as oak trees (genus *Quercus*) and black locusts (*Robinia pseudoacacia*), have been reported in Korea since the 1990s, leading to its classification as a potential forest pest depending on population size [4,5]. In 2001, there was a significant outbreak of *R. mikado* that covered 21,000 hectares of broadleaved forests in the southeastern part of Korea [6], followed by a more recent occurrence in mountain hills around metropolitan areas (Mts. Bong and Cheonggye) that attracted greater public attention due to the high likelihood of encounters with residents [7]. Outbreaks of *R. mikado* have also been reported in the Akashina area of Japan in recent

years [8]. Warm winters are believed to contribute to successful overwintering and subsequent outbreaks of *R. mikado* [7,9]. Foliar damage by stick insects can lead to reduced tree growth and even tree death over successive years [5,10], making it crucial to establish management strategies in preparation for occasional outbreaks. However, ecological information on *R. mikado*, such as phenology, is limited compared to other major forest pests due to the rarity of its outbreaks.

*Ramulus mikado* has a univoltine life cycle. In Korea, the first-instar nymphs of *R. mikado* hatch from overwintered eggs in the leaf litter layer or soils during the spring season and climb up to the host trees [4]. This insect has five to six nymphal stages during the growing season on the host tree [4,9]. Mature adult females drop their eggs on the ground in midsummer, which overwinter and develop into nymphs the following spring. The young nymphal period, especially the first instar, is considered an ideal target stage for controlling *R. mikado*; using sticky traps rolled at the tree base can be effective in preventing severe defoliation (our unpublished observation), as older nymphal stages or adults consume more tree leaves [4]. Targeting younger nymphs also reduces reproductive potential. However, young nymphs, especially first instars, with their small size and protective coloration, are difficult to monitor in forest conditions. Chemical control against the first instar based on empirical emergence dates in the spring may not be effective, as hatching time may vary with the environmental conditions experienced during the long embryonic stage from midsummer to the following spring. Moreover, even if a foliar application is conducted during the emergence time of the first instar, it may not be effective when young nymphs are more abundant on the ground before climbing up to their host tree. Predictive studies on the phenology of *R. mikado*, such as egg hatching and nymphal dispersal of the first instar, would aid in decision-making for optimal management timing [11].

The development rate of *R. mikado* eggs is influenced by temperature, as is the case with other insect species, making it a useful predictor for the phenological prediction of its first-instar nymph [9,12,13]. Studies have shown that the development time of eggs decreases as temperature increases between 15 and 25 °C and that exposure to low temperatures of 5 °C is not effective for embryonic development [12]. Conversely, egg development is slow or inhibited at high temperatures of 30 °C [13]. Lee et al. [9] proposed a linear and nonlinear development model for eggs of the Korean population of *R. mikado* based on the temperature-dependent response of development time. However, the information from these studies is challenging to apply directly to egg phenology prediction for several reasons. First, the development model for eggs is not clear at high temperature ranges, as it was fitted using four developmental rates at operative temperatures and one zero value at a high temperature [9]. Second, the model suggested by Lee et al. [9] was developed using the thermal response of freshly laid eggs to estimate the proper temperature range for mass rearing under indoor conditions. As a result, the time when eggs are first affected by the surrounding temperature under field conditions is unknown. To apply the laboratory-based development model to predict egg emergence in the spring, it is necessary to determine this biofix date [14].

The objective of this study is to establish a reliable degree-day model to predict the hatching of *R. mikado* eggs in natural conditions. To achieve this, we first expanded upon the previous research conducted by Lee et al. [9] by examining the developmental time of *R. mikado* eggs at high temperature ranges. Second, we used a combination of semifield and laboratory experiments to determine the proper biofix date, which is essential for applying the developmental completion model to predict the timing of egg hatching. Finally, the accuracy of the developed egg hatching model for *R. mikado* eggs was evaluated through field monitoring.

## 2. Materials and Methods

### 2.1. Experiment 1: Temperature-Dependent Development

A laboratory colony of *R. mikado* was used to study the development time of eggs at relatively high temperature ranges. The colony was initiated with adults collected from Mt.

Bong in Eunpyeong-gu, Seoul, in August 2021 and reared on clover plants (*Trifolium repens*). The laboratory colony was maintained in controlled conditions with a temperature of $25 \pm 0.5$ °C, a relative humidity (RH) of 60%–70%, and a photoperiod of 16 light (L):8 dark (D) hours. After three generations, eggs laid by adults were collected from the floor of the rearing acryl cage ($60 \times 30 \times 30$ cm) for 20 female adults. The freshly laid eggs (<1 day old) were placed in a weighing dish (Korea Ace Scientific; Seoul, Republic of Korea) onto an insect rearing dish (Cat. No. 310202, SPL; Pocheon, Republic of Korea). Filter paper (Cat. No. WF6-0900, Hwan-gyeong Tec; Seoul, Republic of Korea) at the bottom of the rearing dishes was kept moist with regular sprays. Each dish contained 10 eggs. The experiment involved more than 50 eggs and was conducted at three temperature points (mean $\pm$ S.D.): $23.3 \pm 0.06$ °C, $28.3 \pm 0.27$ °C, and $29.2 \pm 0.27$ °C. These temperatures represented a range from the expected optimal temperature to the potential upper threshold temperature that might inhibit egg development. The experiment was performed in an environmental chamber (DS-8CL, Dasol Scientific; Hwaseong, Republic of Korea) with a 16 L:8 D photoperiod and 60%–70% RH. The egg hatch was monitored daily, and the experiment was continued for 30 consecutive days from the last observed hatch. The temperature and humidity inside the chamber were monitored using HOBO data loggers (U12-012, OnSet Computer Corp.; Bourne, MA, USA) during the experiment.

### 2.2. Experiment 2: Effect of Sampling Time on Development

To investigate the difference in developmental status according to season, we collected eggs from Mt. Cheonggye in both summer and winter. The early developmental stage of eggs (summer eggs) was collected on 10 August 2021, while the overwintering eggs (winter eggs) were collected on 12 December 2021. Using a trowel, we randomly sampled the surface of the soil (1 m $\times$ 1 m) at five locations where *R. mikado* was found in high concentrations in the growing season. The collected soil samples were placed in zip-lock bags and transferred to the laboratory. We then manually separated *R. mikado* eggs from the soil samples in a temperature-controlled room maintained at $15 \pm 0.5$ °C. Due to the difficulty in visually identifying the eggs and the manual nature of the process, the separation took longer than expected. As a result, the summer and winter eggs were exposed to 15 °C for 16 and 21 days, respectively, during the separation process. The separated eggs were then transferred to insect rearing dishes using the same method described in Experiment 1 (see above section), and the hatching of eggs was monitored daily at $25 \pm 0.5$ °C, a relative humidity of 60%–70%, and a 16 L:8 D photoperiod.

### 2.3. Experiment 3: Effect of Chilling Temperature on Development

To determine the effect of low temperatures on the development of *R. mikado* eggs, we conducted an experiment using eggs collected from a laboratory colony on 20 July 2022. These eggs were separated into three groups, each containing 27 eggs. The first group was kept under controlled conditions with a temperature of 25 °C, RH of 60%–70%, and a photoperiod of 16 L:8 D for the entire embryonic development period. The second and third groups were maintained under the same conditions as the first group for 76 days and then transferred to a refrigerator (C053AF, LG Electronics; Seoul, Republic of Korea) controlled at $6.0 \pm 0.51$ °C (mean $\pm$ S.D.) for 20 and 40 days, respectively. The chilling temperature of 6 °C was set at a level similar to the LDT estimated in our study, and below the LDT suggested by Lee et al. [9]. After exposure to 6 °C, the eggs were relocated to the insectary, where the temperature was maintained at $25 \pm 0.5$ °C. All experimental processes were performed with the same method as in Experiments 1 and 2, with the exception of the temperature regimes. The emergence of the first-instar nymphs was monitored daily, and the developmental times across groups were recorded for comparison.

### 2.4. Data Analysis and Phenology Model Development

We combined the results from Experiment 1 with previously reported data from Lee et al. [9] to assess the effect of constant temperatures on egg development time. The

analysis was performed using the Kruskal–Wallis rank sum test, followed by a post-hoc comparison with Bonferroni correction. The development rates (1/day) for each temperature were calculated as the reciprocal of the mean development time and were then fitted to both linear and nonlinear models, incorporating previous data from Lee et al. [9]. The linear model was represented by the equation:

$$r(T) = aT + b \tag{1}$$

where $r(T)$ is the development rate at a certain temperature of $T$ (°C), and $a$ and $b$ are parameters to be estimated. The lower development threshold (LDT) was estimated as $-b/a$, and the thermal constant (degree-days, DD) was estimated as $1/a$ [15,16].

To characterize the thermal response incorporating the nonlinear portion of development times, we attempted to fit the egg development rates at each temperature using several models: Beta, Lactin-2, and Briére-2 [17–19]. However, both Beta and Lactin models failed to converge during parameter estimation. Given that a previous study on the thermal response of *R. mikado* eggs adopted the Briére model [9,19], we chose it as well for consistency. The model is represented by:

$$r(T) = \alpha T(T - T_L)(T_U - T)^{1/\beta} \tag{2}$$

where $\alpha$ is an empirical constant to be estimated and $\beta$ is a parameter determining the shape of the curve in the high temperature range. $T_L$ and $T_U$ represent the LDT and upper development threshold temperature (UDT), respectively. The development rates were fitted, excluding the point of zero development rate (i.e., no hatching) due to the risk of bias estimation in conceptual UDT [20].

To describe the variation in egg development, the cumulative frequency of development completion was calculated. The development times were normalized to the median development time at each temperature to characterize the species' own development distribution, regardless of temperature conditions. The cumulative frequency of development time was fitted against the normalized time using a two-parameter Weibull cumulative function. The model fitting was conducted using previous data examining other constant temperatures from Lee et al. [9].

$$F(x) = 1 - Exp\left[(-x/\gamma)^{\delta}\right] \tag{3}$$

where $F(x)$ represents the cumulative frequency of development completion at the normalized time $x$, and $\alpha$ and $\beta$ are the scale and shape parameters, respectively. The model $F(x)$ was then scaled by multiplying the thermal constant estimated by the linear model by the normalized time. All statistical analyses and model fitting, except for the nonlinear development model, were performed in R 4.1.1 [21]. The fitting of the nonlinear Briére model was conducted using PROC NLIN in SAS 9.4 [22].

To predict the egg hatching of *R. mikado* in field conditions, a development completion model based on the thermal response of fresh eggs of *R. mikado* was applied (Experiment 1). The calendar date when the eggs of *R. mikado* begin to be affected by the surrounding temperature was estimated. Since the process and conditions of summer diapause and its termination are not well understood [13], we empirically evaluated five potential biofix dates: 1 June, 1 July, 1 August, 1 September, and 1 October. This approach is a modification suggested by a previous study that investigated the phenology of a stick insect, *Megacrania tsudai*, in Japan [23]. All calendar dates, up to the collection times for both summer and winter eggs, were transformed into daily degree days. For this transformation, we used the daily maximum and minimum temperatures from the nearest weather stations: Mt. Bong (~0.3 km away) and Mt. Cheonggye (~5 km away) from the sampling site. The single sine wave method with estimated LDT and UDT was applied to calculate the degree days. After the collection date, controlled room temperatures were applied to calculate the daily degree days (i.e., temperature–LDT). The daily degree-days were accumulated from

the potential five dates. The expected emergence date was determined when the accumulated degree days reached the thermal constant derived from the linear development rate model. The estimated emergence dates with five different starting points were compared to the observed median emergence dates. The deviation between these dates was then calculated, and the biofix date with the smallest absolute deviation was selected to forecast the egg hatching of *R. mikado* in field conditions.

The effect of low temperature and its exposure duration was evaluated in Experiment 3 by determining the mean duration times at 25 °C for three groups with different exposure times of 6 °C. Since the data showed normality (Shapiro–Wilk test; $p > 0.05$), the mean durations of these groups were separated using Tukey's HSD test ($p < 0.05$).

*2.5. Model Validation in Field Conditions*

In the spring of 2022, the egg hatching of *R. mikado* was monitored using soil emergence traps (BT2008, BugDorm; Taichung, Taiwan) on Mt. Bong and Mt. Cheonggye (Figure 1 and Table 1). The monitoring sites were classified into three categories: north slope (N), south slope (S), and ridge (R), and the traps were placed on the flat surface of the ground. The appearance of first-instar nymphs of *R. mikado* was checked every week. Additionally, soil samples containing *R. mikado* eggs were transferred to the National Institute of Forest Science (NIFoS) on 23 March 2022 (BO_NIFoS and CG_ NIFoS). The soil samples were placed in acrylic cages (60 × 30 × 30 cm) outdoors at the NIFoS (37°35′37.7″ N 127°02′43.1″ E), and the emergence of the first instars was monitored daily. At three monitoring sites (BO_R-1, CG_R-4, and CG_S-4), the dispersal of first instars was monitored along with egg hatching (Table 1). Five trees around the soil emergence trap were selected, and a sticky trap (RT20-Y100, Daegil Co., Ltd.; Gimhae, Republic of Korea) was rolled up on the tree trunk (Figure 1). Monitoring was conducted until a month after the last appearance or catch of the first instar. The temperature inside the emergence traps was recorded using a HOBO data logger in mid-March.

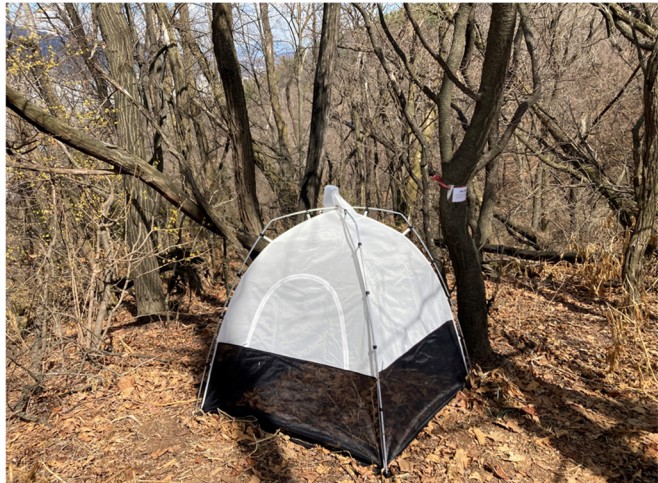 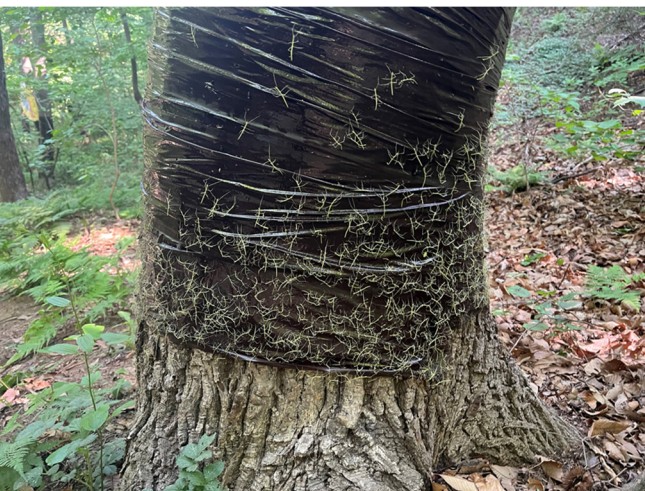

**Figure 1.** Monitoring the appearance and dispersal of first-instar nymphs of *Ramulus mikado* using a soil emergence trap (**left**) and a sticky trap (**right**).

The development completion model was applied to forecast the egg hatching of *R. mikado*. By multiplying the estimated thermal constant by the normalized time, the model was scaled and compared to the field monitoring data. Daily degree days were calculated using the maximum and minimum temperatures, combining weather station data with the recorded air temperature data inside the emergence trap. The LDT and UDT estimated from the linear and nonlinear development rate models were used to compute daily degree days with the single sine wave method. The daily degree days were accumulated from the biofix date, which was empirically determined in this study. Using

the scaled development completion model, the predicted Julian date corresponding to the observed cumulative frequency of nymphal emergence was calculated. The absolute deviation between prediction and observation was then calculated. Due to the asymptotic nature of the model at 0 and 1, cumulative frequencies ranging from 0.05 to 0.95 were considered for the phenology model evaluation. Similarly, the predicted egg hatching was compared with the observed nymphal dispersal, which was monitored by the roll trap, to estimate the time gap between these two phenological events. Using the phenology model for eggs, dates corresponding to the observed cumulative frequency of nymphal dispersal were calculated. These predicted dates were then compared to the actual observed dates for nymphal dispersal and averaged.

**Table 1.** Summary of field monitoring data for emergence and dispersal of first instar of *Ramulus mikado*.

| Sampling Site | Type | Code | No. of Eggs Hatching | No. of Individuals Caught in Sticky Trap |
|---|---|---|---|---|
| Mt. Bong | NIFoS | BO_NIFoS | 24 | - |
| | Ridge | BO_R-1 | 24 | 529 |
| Mt. Cheonggye | NIFoS | CG_NIFoS | 30 | - |
| | North slope | CG_N-1 | 61 | - |
| | | CG_N-2 | 165 | - |
| | | CG_N-3 | 66 | - |
| | Ridge | CG_R-1 | 24 | - |
| | | CG_R-2 | 19 | - |
| | | CG_R-3 | 70 | - |
| | | CG_R-4 | 25 | 2839 |
| | South slope | CG_S-1 | 189 | - |
| | | CG_S-2 | 106 | - |
| | | CG_S-3 | 77 | - |
| | | CG_S-4 | 26 | 7920 |

Note: The 'NIFoS' eggs were transferred to the National Institute of Forest Science (NIFoS) on 23 March 2022.

## 3. Results

### 3.1. Results of Experiment 1: Temperature-Dependent Development

Eggs of *R. mikado* developed at the three experimental temperatures. A small number of eggs could develop at high temperatures of 28.3 and 29.2 °C (Table 2). The results showed that the development time of *R. mikado* eggs was significantly affected by temperature ($\chi^2$ = 115.51, $df$ = 6, $p$ < 0.0001). The development time decreased as the temperature increased, but increased again at the high temperature of 29.2 °C.

**Table 2.** Development time of *Ramulus mikado* at constant temperature.

| Temperature (°C) | Initial Number | Development Time (Days) (Mean ± S.D.) | Development Time (Days) (Median) | Survival Rate (%) | Reference |
|---|---|---|---|---|---|
| 15.4 | 65 | 204.5 ± 4.74 | 206 | 21.5 | [9] |
| 20.3 | 60 | 115.8 ± 6.46 | 114 | 38.3 | |
| 22.9 | 103 | 100.9 ± 4.20 | 100 | 35.0 | |
| 24.9 | 63 | 90.5 ± 6.90 | 88 | 31.7 | |
| 29.3 | - | - | - | - | |
| 23.3 | 58 | 90.6 ± 3.54 | 90 | 63.8 | This study |
| 28.3 | 61 | 81.4 ± 6.83 | 78 | 8.2 | |
| 29.2 | 52 | 123.0 ± 3.27 | 123 | 5.8 | |

The linear portion of the development rate of *R. mikado* eggs could be described by a linear model ($F$ = 70.41; $d.f.$ = 1, 4; $p$ < 0.01) (Table 3 and Figure 2A). The lower developmental threshold (LDT) and thermal constant of the eggs were estimated to be 6.1 °C and 1707.8 DD, respectively. The overall development rate was explained by a nonlinear

model ($F$ = 689.0; *d.f.* = 3, 3; *p* < 0.001) (Table 3 and Figure 2A). The upper developmental threshold (UDT) of *R. mikado* eggs was estimated to be 29.2 °C. The cumulative frequency of development completion was described by the two-parameter Weibull cumulative function, with estimated parameter values of $\gamma$ = 1.008 ± 0.0010 and $\delta$ = 50.870 ± 3.8360 (Figure 2B).

**Table 3.** Estimated parameter values of the linear and nonlinear development rate models for *Ramulus mikado* eggs.

| Model | Parameter | Estimate (Mean ± SEM) | $R^2$ |
|---|---|---|---|
| Linear model | $a$ | $5.86 \times 10^{-4} \pm 6.98 \times 10^{-5}$ | 0.95 |
| | $b$ | $-3.6 \times 10^{-4} \pm 1.60 \times 10^{-3}$ | |
| Nonlinear model | $\alpha$ | $1.6 \times 10^{-5} \pm 2.85 \times 10^{-6}$ | 0.97 |
| | $T_L$ | $0.30 \pm 0.529$ | |
| | $T_H$ | $29.22 \pm 0.036$ | |
| | $\beta$ | $7.82 \pm 2.605$ | |

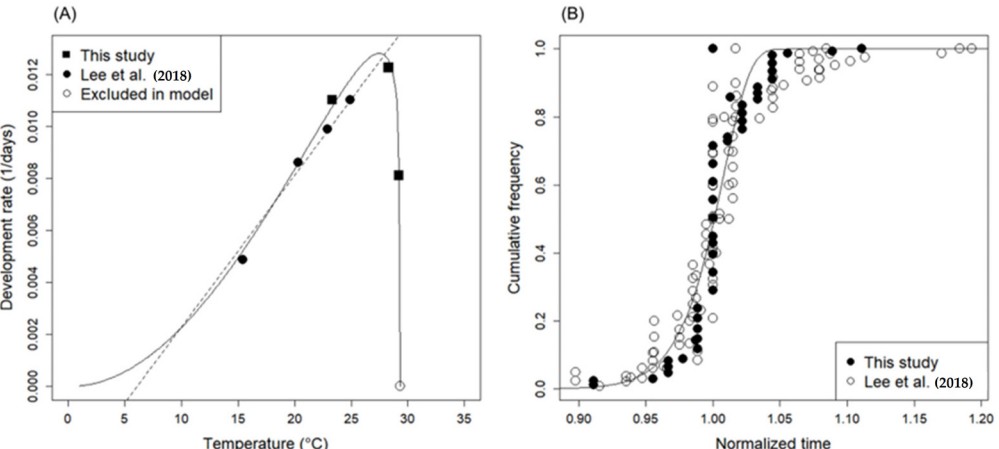

**Figure 2.** (**A**) *Ramulus mikado* egg development rate model as a function of temperature. The dashed line represents a linear model, while the solid line represents a nonlinear model. (**B**) Cumulative distribution model of the development completion frequency of *Ramulus mikado* eggs [9].

### 3.2. Results of Experiment 2: Effect of Sampling Time on Development

Including the separation time of eggs from soil samples in a 15 °C room, the development of the summer eggs of *R. mikado* under laboratory conditions took 82.8 ± 13.36 days. In contrast, the winter eggs required a shorter duration of 29.2 ± 1.63 days after collection from the field. Of the five candidate biofix dates, 1 August yielded the smallest deviation between observation and prediction (Table 4). Therefore, a development completion model scaled by the thermal constant of 1707.8 DD was applied from 1 August to forecast the emergence of the first-instar nymphs of *R. mikado* under field conditions.

### 3.3. Results of Experiment 3: Effect of Chilling Temperature on Development

The exposure of *R. mikado* eggs to low temperatures (6 °C) had no effect on their development time. The results from three groups exposed to different durations of low temperature (0, 20, and 40 days) showed no significant difference in the total time required for hatching at 25 °C ($F$ = 0.69; *d.f.* = 2,12; *p* = 0.52).

### 3.4. Model Validation in Field Conditions

The developed phenology model for the hatching of *R. mikado* eggs showed good agreement between the prediction and the actual emergence that was monitored in the field (Figure 3). The mean absolute deviation between the model predictions and observations was 3.2 ± 2.95 days. The dispersal time of the first-instar nymphs was 6.6 ± 4.76 days later than the egg hatching model.

**Table 4.** Comparison between actual median emergence dates in laboratory conditions of field-collected eggs and the predicted 50% emergence time according to biofix dates.

| Season in Which Eggs Were Collected | Biofix Dates | Date by Which 50% Emergence is Predicted | Observed Median Emergence | Deviation |
|---|---|---|---|---|
| Summer (n = 34) | 1 June | 9 September 2021 | 4 November 2021 | 56 |
| | 1 July | 10 October 2021 | | 31 |
| | 1 August | 6 November 2021 | | −2 |
| | 1 September | 12 December 2021 | | −30 |
| | 1 October | 30 December 2021 | | −56 |
| Winter (n = 83) | 1 June | 2 September 2021 | 19 January 2022 | 139 |
| | 1 July | 7 October 2021 | | 104 |
| | 1 August | 17 January 2022 | | 2 |
| | 1 September | 17 February 2022 | | 29 |
| | 1 October | 10 March 2022 | | 50 |

Note: The biofix date is the starting date for degree-day accumulation. The predicted 50% emergence was the egg hatching time corresponding to the thermal constant estimated by the linear development model. The absolute deviation between the prediction and observation dates was calculated.

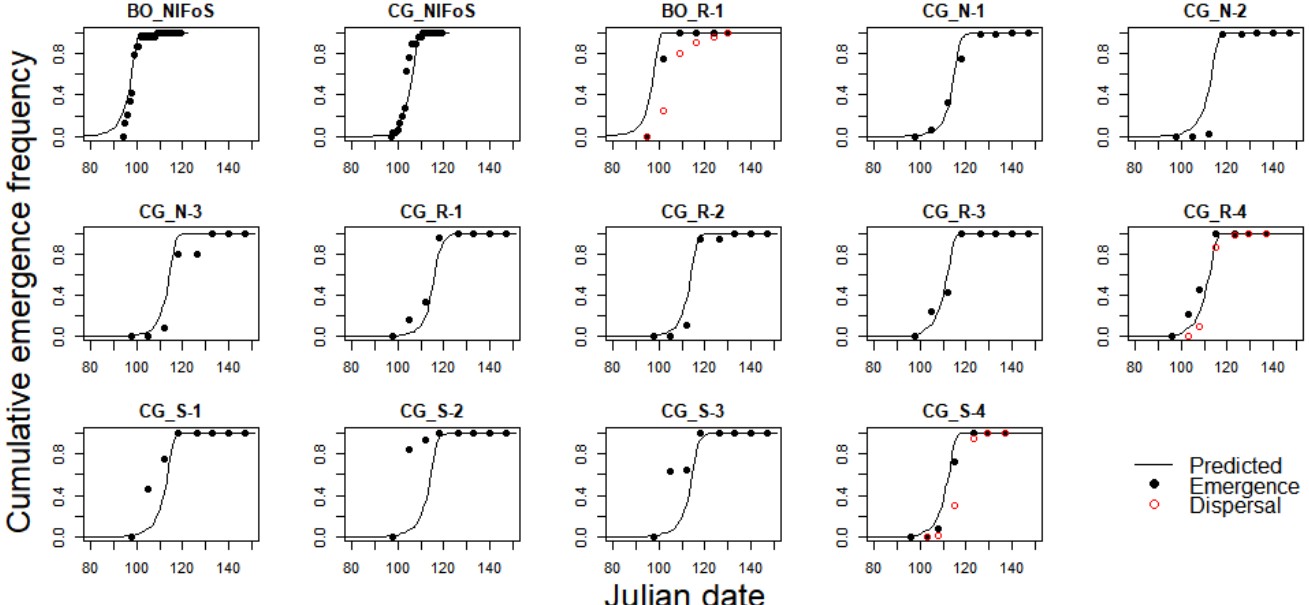

**Figure 3.** Field validations of a phenology model for egg hatching of *Ramulus mikado*. Codes for sampling information at each site are given in Table 1.

## 4. Discussion

The egg hatching time of *R. mikado* plays a critical role in its life cycle success in field conditions. If egg hatching occurs too early in the late summer or fall seasons after oviposition, high temperatures can negatively affect nymphal stages and may result in failure to overwinter [4]. Additionally, young nymphs prefer to feed on soft leaves, such as those of *R. pseudoacacia* and cannot feed on older, thicker leaves, which are more abundant in the summer season (our unpublished observation). Thus, early emergence in the spring season before the growing season of host trees is also not desirable, as the first-instar nymphs can only survive for an average of 6.3 ± 3.18 days without food (our unpublished observation).

The timing of egg hatching in *R. mikado* is likely regulated by high temperatures in the summer during its early development stage and low temperatures in the winter during its pharate development stage. Nakamura and Fukushima [13] suggested that the early embryonic stage of the eggs enters diapause at high temperatures of 30 °C and terminates this phase when exposed to relatively low temperatures. The eggs may therefore slow down or stop developing in response to summer temperatures in Korea. The overwintering

eggs are expected to commence development in response to favorable temperatures in early spring in Korea.

In this study, we extended previous reports on the thermal response of *R. mikado* eggs by conducting developmental experiments at high temperature ranges. The new linear development rate model indicates that this insect may start to be affected by lower temperatures, with an LDT of 6.1 °C, compared to the previously reported 7.6 °C. Additionally, the expanded datasets generated a different nonlinear response compared to the previous study, showing a much higher developmental rate at 27.5 °C and a more rapid decline after this temperature. Although the phenology model developed in this study did not consider the dynamic process of egg diapause, it is expected that the model partially reflects the inhibition of development by the temperature-dependent developmental response. However, further study of the diapause process is needed to improve the egg phenology model, which will provide a more rigorous prediction of the response to diverse environmental conditions.

A significant finding from this study was the difference in developmental status between summer and winter eggs of *R. mikado*. Although we were unable to fully control the separation time of eggs from soil samples at 15 °C, the development time of winter eggs was approximately 50 days faster than that of summer eggs. This suggests that winter eggs collected in December may have already undergone embryonic development before the cold season and entered a quiescence phase, which is a facultative arrest of developmental responses to harsh conditions [24,25]. These results align with experimental evidence that low temperatures (≤6 °C) do not significantly affect egg development, but development resumes once the eggs are exposed to appropriate temperatures again [12]. This highlights the importance of determining the proper starting point for a phenology model to predict the emergence time of field populations.

Based on the development experiments with field-collected populations, we estimated 1 August as the optimal biofix date for forecasting egg hatching in field conditions. This biofix date produced the lowest error in the prediction of the median emergence time of both summer and winter eggs and stable performance in field validation. Park et al. [4] reported that the oviposition period for *R. mikado* occurs in mid-July to early August, with an increase in the proportion of mature eggs in adult ovaries starting from mid-July. While this may not precisely reflect the start of development time for each individual egg in the field, it is roughly in line with our estimated biofix date of 1 August. The high temperatures in the Korean summer may inhibit or slow egg development, as indicated by the nonlinear development rate model. Hence, the effect of oviposition time distribution on individual variation in development may be offset by the summer temperature. In this study, we observed a dramatic decrease in the variation of winter egg development compared to that of summer eggs, which suggests that the development status of eggs in the field may be synchronized by the temperatures they experience from mid-summer to winter. Based on these results, we suggest using the biofix date of 1 August as a general indicator for predicting the nymphal emergence of *R. mikado* in the following spring season, although it may not account for individual variations in development time in the early phase of egg development. However, since the biofix date in this study was empirically estimated without considering the diapause and its termination of this stick insect, it would be necessary to investigate the dynamics of the egg diapause process further for more rigorous forecasting in egg hatch. Factors such as humidity or light cycle, which could influence diapause, should also be considered with additional monitoring data [13,24].

The mean egg development time for *R. mikado* was relatively long, ranging from 81.4 to 204.5 days, when compared to other insect species (Table 2). This aligns with earlier reports suggesting that stick insects generally need an extended egg phase, regardless of diapause [23,26]. Therefore, the extended embryonic development during the egg period in *R. mikado* is a typical characteristic of the Phasmatidae family. However, the thermal response of *R. mikado* differs from that of the subtropical stick insect, *M. tsudai*. Specifically, *M. tsudai* showed a faster development rate at a high temperature of 30 °C compared

to 20 and 25 °C. This may be a result of adaptative evolution in response to the higher temperature of subtropical regions, where the plant growing season persists throughout the year. This adaptation could explain the longer appearance season of first-instar nymphs of *M. tsudai*, from October to the following April, in its native range. In contrast, *R. mikado* does not occur during the winter season in Korea [4]. This is likely modulated by a diapause or quiescence phase coupled with a low development rate in summer and a high thermal constant, preventing unfavorable emergence in the fall before the onset of the cold winter. As such, a summer developmental arrest or inhibition due to high temperatures might be a crucial component of the life cycle of this insect in temperate regions.

When the results of low or high temperatures followed by operational temperature treatments were transformed into cumulative degree-days, over half of the experimental results showed small differences within 100 DD of the thermal constant (1707.8 DD) estimated in this study (Table S1). However, for some results, particularly those performed on the Japanese population, a relatively larger variation was observed. This variation may be attributed to population-specific characteristics, such as differences in thermal response, diapause intensity, or local adaptation [13]. The lower cumulative degree days in the population exposed to 30 °C could be due to the limitations of the development rate model in fully explaining the response to high temperatures (Table S1). The development rate at a given temperature is theoretically calculated based on the reciprocal of the time required for development completion. However, the development time at approximately 30 °C was not determined (no hatch), leading to a zero-development rate and no accumulation of daily degree-days. Despite this, longer exposure to 30 °C was found to slightly shorten the egg development time at the operational temperature [13]. Hence, extended periods of high temperatures of approximately 30 °C, which are not common in Korea, may contribute to certain stages of egg development. However, this is not enough to complete egg hatching due to slow developmental rates or the incomplete termination of summer diapause. To comprehensively grasp the thermal response of *R. mikado*, future studies should intensively investigate the impacts of rising temperatures, linked to climate change, on summer diapause development.

The developed phenology model for *R. mikado* eggs was able to accurately predict the completion of first-instar emergence under different environmental conditions. This makes it a useful tool for determining optimal sampling times for young *R. mikado* nymphs in the field, which aids in management decision-making. Monitoring nymphal density using sticky traps rolled around the tree trunk, based on the model's predictions, would be a good indicator for informed management decisions. Given the 6.6-day time lag between the estimated nymphal appearance and their movement to host trees, the most appropriate time for foliar application would likely be from one week after the predicted 100% emergence date of the first instar. However, foliar insecticide spray and mass trapping by sticky traps can have unintended negative impacts on nontarget organisms, making them undesirable management options for forest ecosystems. Therefore, alternative management options, such as host-specific micro-organism biopesticides, need to be developed to control *R. mikado* outbreaks, and traditional management methods, including foliar spray and trapping, should be used only when necessary and with careful monitoring [27].

## 5. Conclusions

In summary, we studied the development time of *R. mikado* eggs across varied temperature regimes to predict their field emergence time. We identified the operative temperature range, as well as the lower and upper threshold temperatures, for this stick insect's egg development. The phenology model we developed can guide effective sampling and inform forest management decisions. Our findings provide new insights on this potential forest pest, previously understudied in ecological research. However, research on the factors of outbreak and the developmental dynamics of eggs will be necessary in the future.

**Supplementary Materials:** The following supporting information can be downloaded at: https://www.mdpi.com/article/10.3390/f14091710/s1, Table S1: The mean development time (days) of *Ramulus mikado* eggs observed in a combined treatment of low or high temperature followed by an operative temperature. The experiments were conducted under a 16 L:8 D photoperiod condition.

**Author Contributions:** Conceptualization, M.-J.K., J.-K.J. and Y.N.; methodology, M.-J.K., J.-K.J. and Y.N.; software, M.-J.K.; validation, M.-J.K., K.E.K. and S.S.; formal analysis, M.-J.K. and K.E.K.; investigation, M.-J.K., K.E.K. and S.S.; resources, S.S., J.-K.J., Y.P. and Y.N.; data curation, M.-J.K., Y.P. and Y.N.; writing—original draft preparation, M.-J.K.; writing—review and editing, M.-J.K., Y.P., J.-K.J. and Y.N.; visualization, M.-J.K.; supervision, J.-K.J. and Y.N.; project administration, Y.N. All authors have read and agreed to the published version of the manuscript.

**Funding:** This study was supported by the National Institute of Forest Science (project No. FE0703-2022-01), Republic of Korea.

**Data Availability Statement:** The data presented in this study are available upon request from the corresponding author. The data are not publicly available due to institutional policy.

**Acknowledgments:** We are very thankful to Jingu Lee of Gyeonggi Agricultural Research and Extension Services, Republic of Korea, for his advice on rearing insects and for sharing data from development experiments.

**Conflicts of Interest:** The authors declare no conflict of interest.

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
