# Peer review of "Development of a Phenology Model for Egg Hatching of Walking-Stick Insect, Ramulus mikado (Phasmatodea: Phasmatidae) in Korea"

_forests, doi:10.3390/f14091710_

Round 1
Reviewer 1 Report
This paper is relatively well written and provides useful information on this insect. The methods could use some further clarification and two years of field data or data from field sites tat had different temperature profiles would have been better. Discussing how their data on this insects compares to that for other insects may shed some light on the diapause or quiescence that this insect goes through.
Line 97 should specify that eggs were being evaluated.
Lines 106-108 These temperature treatments are very precise. Please provide the standard error on the actual temperatures the eggs were exposed to. Also, give some indication of the variation in temperature for subsequent temperature regimes used in the experiments.
Line 139 Why was 6°C chosen for this experiment? Was it below the developmental threshold for the eggs?
Line 147-148 was the data checked for normality? If the data was not normally distributed then a different type of distribution and or model would be needed. For example if the data had overdispersion then in SAS you would use a GLIMMIX model with possibly a gamma distribution and log link function.
Line 159-168 Were other non-linear models tried? Why was the Briere model chosen?
Lines 200-202 How was fit of the model evaluated. One normally used measure is the AIC. Please provide model evaluation and fit statistics information.
Lines 209-224 In this field work were there differences in the recorded temperatures between aspects? Were there difference in timing of hatch due to temperature differences? It would have been better to have monitored multiple sites that were known to different in temperature regimes or to monitor over multiple years.
Lines 239-241 How was the difference in the estimated and observed calculated? Was each time frame compared and averages calculated?
Line 272-274 Was the time spent at 15 C during sorting included in this estimate? That time is above the developmental threshold and should be incorporated in some way.
Table 4 It is possible that the biofix may differ between years if there are aspects of the diapause that are still not known. It seems that the timing of diapause start and what conditions are needed for diapause termination are still not known. Your data showed no effect of 6C on egg hatch so it is possible that some other factor like humidity or light cycle may be involved. It is also possible that there is no diapause and it is just quiescence. How does the temperature regime for the year this was evaluated compare to other years? Generally, it is better to have more than one year of field data to base this type of estimate on. Is there other information on first or peak hatch form previous observations in other years that could be included?
Discussion: How does what you found for this insect compare to what has been published for other insects? Are there other insects that have similar egg temperature responses? How can that help you better understand what is happening with this insect?
There are a few minor English issues that should be dealt with.
Author Response
We appreciate the reviewers for thoughtful and constructive comments on our manuscript. We have incorporated detailed information regarding our experiments and made revisions in line with the comments provided. All changes in the revised manuscript are highlighted in red for clarity. We believe these revisions have greatly enhanced the quality of our manuscript. We appreciate the time and effort you have invested in examining our work.
Please see the attachment.
Best regards,

Reviewer 2 Report
The captivating and detailed introduction of the study serves as a true gateway into the subject, guiding the reader through the context and complexity of the research in a documented and easily understandable manner.
Material and Methods
The article presents a well-defined and detailed methodology for each of the three experiments, providing readers with a clear understanding of how the study was conducted.
The details regarding environmental conditions, temperature, and other experimental aspects contribute to the reproducibility of the study, which is essential in research.
Suggestions for improving this section:
Line 101, sentence: "16 L:8 D hours." I suggest avoiding the use of acronyms as there might be readers who don't understand that this is an acronym for "light-dark."(just for the first time when you use the term)
In section 2.3. Data analysis and development of the phenology model, it is specified that ANOVA and the Tukey test were used. This implies that the data were previously tested for normality. I suggest specifying whether this normality testing of data was performed.
Line 193 - It is mentioned that climatic data from the nearest weather station were used. I suggest specifying the respective weather stations as well as the distance from them.
The results are presented in a clear and detailed manner. The significant differences in egg development based on temperature and time are highlighted, and details about the observed trends are provided. Additionally, the inclusion of models to explain these results is a positive aspect, indicating an analytical approach and scientific reasoning.
Recommendations for Technical Editing:
Thematic paragraph organization: First, divide the Results section into distinct paragraphs for each experiment (Experiment 1, Experiment 2, Experiment 3). This will make it easier for readers to follow the results for each individual experiment.
Use of subtitles for experiments: Add subtitles for each experiment to mark the beginning of each section. For example, "Results of Experiment 1: Temperature-dependent," and so on.
The discussions are well-addressed with a comprehensive analysis of the results in comparison to other studies.
I suggest placing more emphasis on two aspects:
Limitations and future directions: When discussing the limitations of the study and future directions, you can provide specific examples of how these aspects could be addressed. For instance: "In the future, it could be valuable to delve deeper into the dynamics of the egg diapause process. This could contribute to improving the accuracy of our phenology model and better predicting the response to varied environmental conditions."
Reiteration of impact: In the conclusion of the discussion section, you can reiterate the importance of your findings. For example: "In conclusion, this study makes significant contributions to the understanding of R. mikado egg development and the influence of environmental factors. The developed phenology model proves to be a valuable tool for population management and the ecological approach to the life cycle of this species."
Author Response

(The authors gave the same response as above.)

Round 2
Reviewer 1 Report
The authors have dealt with the major issues. My only suggestion would be that they could include the other non-linear models that were attempted in the methods and state that they failed to converge.
Author Response
Dear Editor and Reviewers,
We are grateful for your time and effort dedicated towards enhancing the quality of our work. We believe the manuscript has significantly improved since the initial version, thanks to your comments.
Best regards,
Response to Reviewer 1 Comments
General comment
The authors have dealt with the major issues. My only suggestion would be that they could include the other non-linear models that were attempted in the methods and state that they failed to converge.
Our response
Thank you for your positive feedback on the revised manuscript. We have now incorporated information about our attempts with other non-linear models in the Materials and Methods section, as you suggested. We appreciate the time and effort you invested in reviewing our work.